environmental chemistry/atmospheric chemistry/environmental science

WFGD, synergistic dust removal efficiency, sieve-tray, droplets swarm model, foam layer

**Author for correspondence:**
Min Gu
e-mail: mgu@cqu.edu.cn

# Synergistic removal of dust using the wet flue gas desulfurization systems

Qirong Wu[1,2], Min Gu[1], Yungui Du[2] and Hanxiao Zeng[1]

[1]State Key Laboratory of Coal Mine Disaster Dynamics and Control, College of Resources and Environmental Science, Chongqing University, Chongqing 400044, People's Republic of China
[2]SPIC (State Power Investment Corporation) Yuanda Environmental Protection Engineering Co., Ltd., Chongqing 400012, People's Republic of China

QW, 0000-0001-8878-0918

Coal is still a major energy source, mostly used in power plants. However, the coal combustion emits harmful $SO_2$ and fly ash. Wet flue gas desulfurization (WFGD) technology is extensively used to control $SO_2$ emissions in power plants. However, only limited studies have investigated the synergistic dust removal by the WFGD system. Spray scrubbers and sieve-tray spray scrubbers are often used in WFGD systems to improve the $SO_2$ removal efficiency. In this study, the synergistic dust removal of WFGD systems for a spray scrubber and a sieve-tray spray scrubber was investigated using the experimental and modelling approaches, respectively. For the spray scrubber, the influence of parameters, including dust particle diameters and inlet concentrations of dust particles, and the flow rates of flue gas and slurry of limestone/gypsum on the dust removal efficiency, was investigated. For the sieve-tray spray scrubber, the influence of parameters such as the pore diameter and porosity of sieve trays on the dust removal efficiency was examined. The study found that the dust removal efficiency in the sieve-tray spray scrubber was approximately 1.1–10.6% higher than that of the spray scrubber for the same experimental conditions. Based on the parameters investigated and geometric parameters of a scrubber, a novel droplets swarm model for dust removal efficiency was developed from the single droplet model. The enhanced dust removal efficiency of sieve tray was expressed by introducing a strength coefficient to an inertial collision model. The dust removal efficiency model for the sieve-tray spray scrubber was developed by combining the droplets swarm model for the spray scrubber with the modified inertial collision model for the sieve tray. The results simulated using both models are consistent with the experimental data obtained.

# 1. Introduction

More than 25% of the primary energy is produced through coal combustion, and coal-fired power plants with a capacity of 1995 GW are producing energy globally until January 2018 [1]. The air pollutants released from coal-fired power plants including acid gases and fly ash could have adverse human and ecosystem health consequences. Hence, the emissions of coal-fired power plants are regulated in many countries [2]. The Chinese standard is the most stringent, and it stipulates that the coal-fired power plants should be retrofitted to reduce the fly ash and $SO_2$ below 10 mg $Nm^{-3}$ (even 5 mg $Nm^{-3}$ for some regions) and 35 mg $Nm^{-3}$ to a reference oxygen content of 6%, respectively, before 2020 [3].

The wet flue gas desulfurization (WFGD) technology is the most commonly used technology for controlling $SO_2$ emissions [4] and its removal efficiency could be improved by installing some strengthened devices [5,6] such as inserting sieve trays in a spray scrubber. The capacity of power plants that use the WFGD systems and the sieve-tray spray scrubbers has exceeded 100 GW since 2014. Most past studies on the WFGD systems focused on the desulfurization efficiency of sieve trays [7–11], the desulfurization mechanisms and mass transfer models for desulfurization [12–14]. However, only limited studies have investigated the dust removal potential of the WFGD system [15,16].

Particulate matter (PM), which is the dust particle with an aerodynamic diameter less than 100 μm, is inhalable and could cause significant negative health impacts on organisms [17]. Kim et al. [18] found that the wet scrubber could sufficiently remove PMs smaller than 1.0 μm under the optimum operational conditions. Furthermore, the PM removal by the WFGD technology is less expensive when compared with that of the wet electrostatic precipitator [19]; hence, it has been used in power plants. Wei et al. [20] reported that the particles removal efficiency of WFGD systems increases with the increase of particle size. Almost all of the particles with the aerodynamic diameters larger than 50 μm (PM50) could be removed by a WFGD system [21], while the particles removal efficiencies were 28.7% for PM1 and of 39.6% for PM2.5 in a 1000-MW power plant [22]. Although the dust concentration of 5 mg $Nm^{-3}$ at the WFGD outlet was difficult to achieve when the inlet dust concentration was above 20 mg $Nm^{-3}$ [19,23], the outlet dust concentration of 5 mg $Nm^{-3}$ was achieved for some power plants [24,25].

In addition to the particle size and dust concentration, the gas flow rate and liquid flow rate could affect the dust removal efficiency in a twin-fluid atomization spray scrubber [17]. In a multiple sieve-plate column, the dedusting efficiency is influenced by the foam density [26,27]. The foam density is a key parameter for the sieve-plate column, which mainly depends on the pore diameter and porosity of the porous sieve tray [7]. The WFGD system generally operates with adjustable operational conditions, while the sieve tray is often used to enhance the efficiency. However, studies on the removal of particles in a WFGD system by considering the inlet dust particle parameters, operational parameters and the spray scrubber with or without sieve tray are limited.

Developing a model to determine the synergistic dust removal efficiency of the WFGD system is important for the engineering application. The basic mechanisms of wet dedusting include inertial collision, diffusion, interception and gravitational settling, among which inertial collision is often dominant, and the diffusion mechanism is important for dedusting of small particles [17]. Consequently, the dedusting mathematical models such as dispersed droplets swarm models for spray scrubbers were established based on these mechanisms [28,29]. In a spray scrubber of the WFGD system, the dedusting efficiency depends on the droplet size of the sprayer, gas flow rate and liquid flow rate [17], and the efficiency was high for a larger liquid-to-gas flow ratio [18]. Furthermore, the dedusting efficiency is enhanced by the sieve tray [5,21,23]. Most studies focused on the dust removal efficiency for either multilayer sieve plate [26,30,31] or spray layers without considering the sieve tray [17,18,32,33]. The dust collection in a sieve-tray spray scrubber was contributed by the dispersed droplets from spray layers, along with the foam layer on the sieve tray. Therefore, a comprehensive mathematical model considering both of them could better describe the synergistic dust removal efficiency of the WFGD system with a sieve-tray spray scrubber.

# 2. Mathematical modelling of dust removal mechanisms

## 2.1. Dust removal model for spray scrubber

### 2.1.1. Dust removal efficiency of a single droplet

The wet dust removal depends on the capture capacity of droplets, which is mainly governed by the inertial collision and diffusion mechanisms [33]. Taheri & Sheih [34] developed a mathematical model

to predict the particle collection efficiency of an atomizing scrubber. The dust collection efficiency ($\eta_I$) contributed by inertial collision is expressed by equation (2.1), where the inertial collision mechanism is defined by the parameter $K_P$ (equation (2.2)):

$$\eta_I = \left(\frac{K_P}{K_P + 0.7}\right)^2 \tag{2.1}$$

and

$$K_P = \frac{C_I d^2 \rho_p u}{9\,\mu_g d_d}. \tag{2.2}$$

Kim *et al.* [18] developed a model to predict the particle removal efficiency considering the diffusion, interception and impaction mechanisms.

$$\eta_D = \frac{4}{P_e}2 + 0.557\,R_{eD}^{\frac{1}{2}}S_c^{\frac{3}{8}} \tag{2.3}$$

where

$$P_e = \frac{d_d u}{D}, \tag{2.4}$$

$$D = \frac{kTC_D}{3\pi\mu_g d}, \tag{2.5}$$

$$R_{eD} = \frac{d_d u \rho_g}{\mu_g}, \tag{2.6}$$

$$S_c = \frac{P_e}{R_{eD}}, \tag{2.7}$$

$$\eta_R = (1 + R)^2 - \frac{1}{1 + R} \tag{2.8}$$

and

$$R = \frac{d}{d_d}. \tag{2.9}$$

The gravitational settlement mechanism is prominent for the removal of big particles. The removal efficiency of gravity settlement $\eta_G$ is given as [35]

$$\eta_G = \frac{C_G d^2 g}{18\,\mu_g \mu_l}. \tag{2.10}$$

Mohan *et al.* [17] assumed that the collection efficiency of a single droplet is the sum of diffusion efficiency ($\eta_D$ equation (2.3–(2.7)) [28], inertial collision ($\eta_I$ equation (2.1)) and interception ($\eta_R$ equation (2.8)–(2.9)) [28,36]. Considering the independence of the contribution of $\eta_I$, $\eta_G$, $\eta_D$, and $\eta_R$, the overall efficiency of a single droplet $\eta_P$ is given as [28,36]

$$\eta_P = 1 - (1 - \eta_I)(1 - \eta_R)(1 - \eta_D)(1 - \eta_G). \tag{2.11}$$

### 2.1.2. Dust removal efficiency of droplets swarm

The overall dust removal efficiency is calculated based on the dispersed droplets swarm in the whole scrubber. Based on the single droplet model, the different swarm droplets model was developed to describe the overall dedusting efficiency of the WFGD system [28]. According to the discrete material balance equation for the dust in a cell volume, the dust removal efficiency $\eta_{SP}$ is given by equation (2.12) [37] and equation (2.13) [28]:

$$\eta_{SP} = 1 - \exp\left(-\frac{3\eta_P \times H \times V_L \times u}{2 \times V_G \times d_d \times (u - u_g)}\right) \tag{2.12}$$

and

$$\eta_{SP} = 1 - \exp\left(-\frac{0.0015\eta_P \times H \times u_g \times V_L}{d_d \times u}\right). \tag{2.13}$$

## 2.2. Dust removal model for the foam layer

For the foam layer, the dust removal efficiency is mainly dependent on the inertial collision ($\eta_{SI}$) and diffusion impact ($\eta_{SD}$), which are given as equation (2.14) and equation (2.18), respectively [32,36]:

$$\eta_{SI} = 1 - \exp(-C_f \cdot S_{tb}), \tag{2.14}$$

where

$$S_{tb} = \frac{C_s d^2 \rho_p u_h}{18\, \mu_g d_h}, \tag{2.15}$$

$C_f$ is the function of mean foam density ($F$), which is determined using equation (2.16) [26],

$$C_f = 40F^2. \tag{2.16}$$

Taheri [26] reported that the foam density was 0.38–0.65 for the sieve-plate column without providing the calculation model. In this article, the foam density was given by equation (2.17) [7]:

$$F = \frac{u_h^{0.28} \rho_g^{0.14}}{\varphi_0^{0.14} \sigma^{0.07} \rho_L^{0.07}} \tag{2.17}$$

and

$$\eta_{SD} = 1 - \exp\left(-\frac{6\,h_b}{\pi} \cdot \left(\frac{3D}{\pi r_b^2 u_b}\right)^{1/2}\right), \tag{2.18}$$

where the height of the foam layer $h_b$ and bubble radius $r_b$ are given by equations (2.19) and (2.20), respectively.

$$h_b = 0.23\left(\frac{V_L}{V_G}\right)^{0.45} (\varphi_0^2 d_h)^{-0.55} \tag{2.19}$$

and

$$r_b = 0.355 R_{eb}^{-0.05}. \tag{2.20}$$

The Reynolds number of sieve tray is given by

$$R_{eb} = \frac{d_h u_h \rho_g}{\mu_g}. \tag{2.21}$$

The rising velocity of bubble $u_b$ is given as [31]

$$u_b = 0.71(g d_b)^{0.5} + u_g, \tag{2.22}$$

where

$$d_b = 0.615 g^{-0.2} d_s^{0.8}\left(u_g \frac{1}{\varphi_0}\right)^{0.4}. \tag{2.23}$$

The total dust removal efficiency $\eta_S$ is given by equation (2.24) [36]:

$$\eta_S = 1 - (1 - \eta_{SI})(1 - \eta_{SD}). \tag{2.24}$$

# 3. Material and methods

## 3.1. Experimental apparatus

The schematic diagram of the experimental apparatus is shown in figure 1. The scrubber consisted of a changeable sieve plate and two spray layers. The mist eliminator was installed on the top of the scrubber. The height ($H$) and inner diameter ($R_t$) of the spray scrubber were 2.0 m and 0.15 m, respectively.

The simulated dust was fed at the ash feeding (AF) point and entered into the scrubber through the compressed air. The pH of the limestone/gypsum slurry was maintained at 6.0. The concentration of the limestone/gypsum slurry was adjusted by the valve at the bottom of the tank (ST).

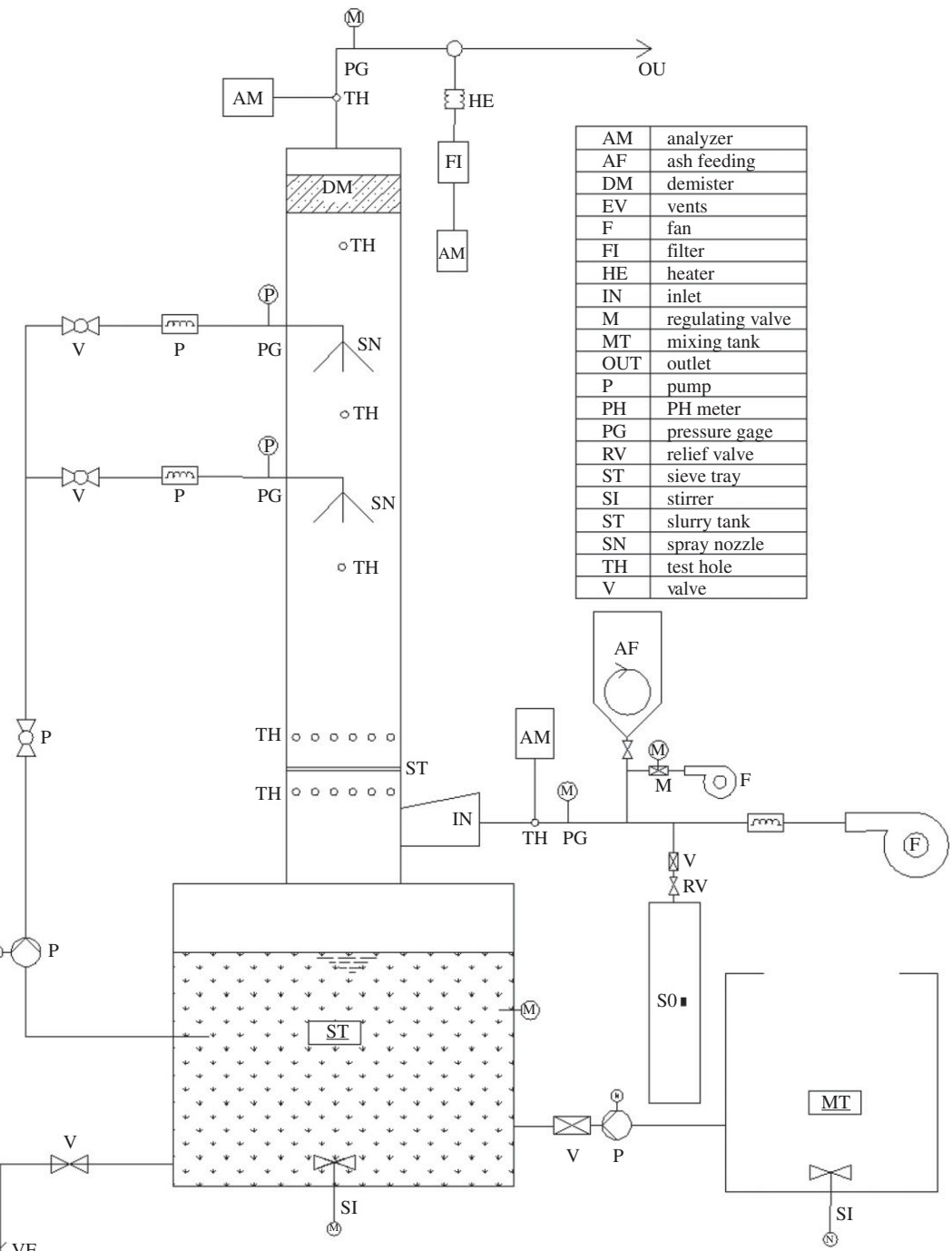

**Figure 1.** Experimental apparatus of the sieve-tray spray scrubber (AM, Analyser; AF, Ash Feeding; DM, Demister; EV, Vents; F, Fan; FI, Filter; HE, Heater; IN, Inlet; M, Regulating Valve; MT, Mixing Tank; OUT, Outlet; P, Pump; PH, PH Meter; PG, Pressure Gage; RV, Relief Valve; ST, Sieve Tray; SI, Stirrer; ST, Slurry Tank; SN, Spray Nozzle; TH, Test Hole; V, Valve).

## 3.2. Material and methods

$SiO_2$ particles with the diameters of 1, 5, 10, 20 and 50 μm (Donghai Mineral Products Co., LTD.) were used to simulate fly ash in the experiments. Ten sieve trays with a thickness of 3 mm were used. The pore diameters of the sieve trays were 5, 10, 15, 25 and 35 mm with the porosities of $30 \pm 1\%$, and the porosities of 21.2%, 25.8%, 32.97%, 35.32% and 40.82% with a consistent pore diameter of 15 mm, respectively.

The simulated flue gas was prepared by mixing $SO_2$ from the gas cylinder ($SO_2 > 99\%$) with the air from the compressor in the pipe. The $SO_2$ concentration was measured by Testo 350 portable flue gas analysers placed in the test hole (TH).

The dust concentration was measured according to Method 5 of American EPA [38]. The ultra-fine glass fibre filter papers (British Whatman Company) were used to collect dust. The filter paper was weighted after drying at 105°C for 2 h before and after collecting the dust. The dust removal efficiency ($\eta$) was calculated by equation (3.1):

$$\eta = \frac{C_0 - C_{out}}{C_0} = \frac{M_I V_I - M_D V_L}{M_I V_I},$$

(3.1)

where $C_0$ (mg m$^{-3}$) is the dust concentration at inlet, $C_{out}$ (mg m$^{-3}$) is the dust concentration at outlet, $M_I$ (mg) is mass of the dust, $V_I$ (m$^3$) is the sample volume and $M_D$ (mg) is the mass difference before and after collecting dust.

# 4. Results and discussions

## 4.1. Experimental and modelling results of dust removal efficiency for spray scrubber

### 4.1.1. Experimental results

#### 4.1.1.1. Effect of dust parameters

The synergistic dust removal efficiency of the WFGD system mainly depends on the dust particle properties in the flue gas such as inlet particle concentrations and particle diameter, and the operating parameters of the system such as flow rates of flue gas and of limestone/gypsum slurry. The effects of dust diameter ($d$) and inlet concentration ($c_0$) on dust removal efficiency ($\eta_{SP}$) of the spray scrubber are shown in figure 2a,b, respectively.

The dust removal efficiency steadily increases with the increase in dust diameter (figure 2a). The dust removal efficiency was only 2.9% for PM1, although it significantly increased to 91.9% for PM20, while $\eta_{SP}$ was close to 100% for PM50. These results are similar to other wet methods that could easily remove large particles, while ineffective in the removal of small particles [17,26,39]. The difference is that the efficiency for every particle's size fluctuates with the column type and operational conditions.

When $c_0$ was less than 99 mg m$^{-3}$, $\eta_{SP}$ decreased slightly with the increase of $c_0$ (figure 2b). $\eta_{SP}$ in the experiment was between 88.9% and 91.7%, which is higher than the industrial test results of about 70% for different commercial power plants [19] because the inlet dust diameter of industrial tests is non-uniform, many particles smaller than 20 µm are contained in the inlet dust, resulting in a lower efficiency. Furthermore, the outlet dust concentrations were 2.61 mg m$^{-3}$ and 6.6 mg m$^{-3}$ for the inlet concentrations of 29 mg m$^{-3}$ and 59 mg m$^{-3}$, respectively. Consequently, achieving an outlet concentration below 5 mg Nm$^{-3}$ for an inlet concentration above 30 mg Nm$^{-3}$ would be difficult, while it would be possible for an inlet concentration below 20 mg Nm$^{-3}$ [23], because the number of droplets was constant when $V_L$ was constant, whereas the droplet–particle collision probability reduced with increased $c_0$.

Moreover, $\eta_{SP}$ declined sharply when the dust concentration was larger than 99 mg m$^{-3}$ because it exceeded the scrubber absorption capacity. The scrubber absorption capacity can be evaluated by the correlation ratio of particles to droplets $n_r$ (equation (4.1)) as follows:

$$n_r = \frac{\text{number of dust particles}}{\text{number of dust droplets}} = \frac{V_G \times c_0 \times d^3}{V_L \times d_d^3 \times \rho_p \times 10^6},$$

(4.1)

$n_r$ is 2.73 when $V_L$ is 1.5 m$^3$h$^{-1}$, Sauter mean diameter ($d$) is 1900 µm, $c_0$ is 99 mg m$^{-3}$, and $V_G$ is 105 m$^3$ h$^{-1}$. Hence, the spray scrubber is suited for dedusting only when the $n_r$ is less than 2.73.

#### 4.1.1.2. Effect of operational parameters

The flow rate of flue gas ($V_G$) and the flow rate of limestone/gypsum slurry ($V_L$) have a significant influence on the dust removal efficiency of the WFGD system as shown in figure 3a,b, respectively, for 20 µm particle. The dust removal efficiency increased from 90.7% to 95.6% when the flow rate of flue gas increased from 79 to 129 m$^3$ h$^{-1}$ (figure 3a). This is because the particle impaction frequency is enhanced when the gas flow increases.

Figure 3b illustrates the increase in the overall efficiency with the increase in liquid flow rates, because of the increase in the number of droplets. The same pattern was observed in the spray scrubber using twin-fluid atomization [17]. For the experimental liquid flow rates of 1.32 m$^3$h$^{-1}$ to 2.26 m$^3$ h$^{-1}$, $n_r$ values are 1.1–2.1, respectively, which are smaller than the critical value of 2.73. Therefore, one dust particle can be captured by more droplets and thus can be easily removed.

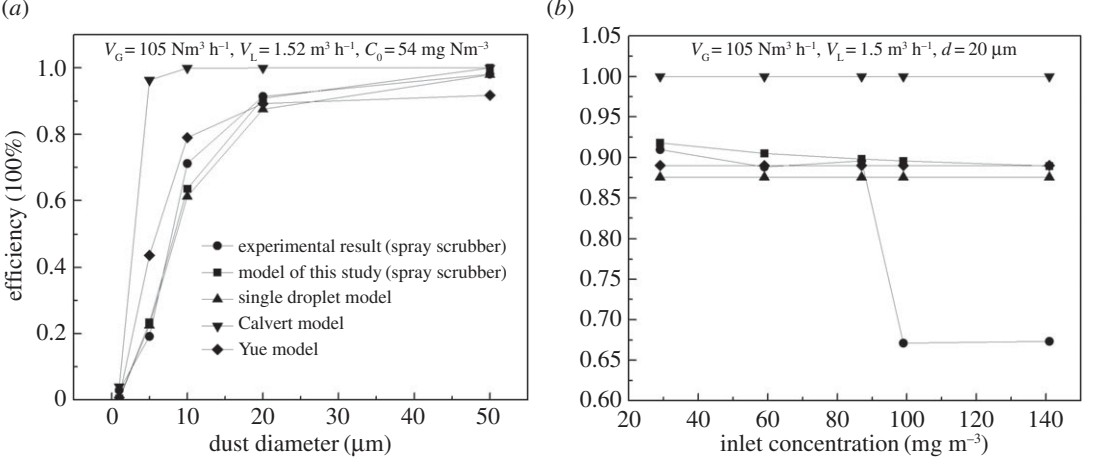

**Figure 2.** Effect of particle parameters on the dust removal efficiency in spray scrubber: (a) dust diameter and (b) inlet dust concentration.

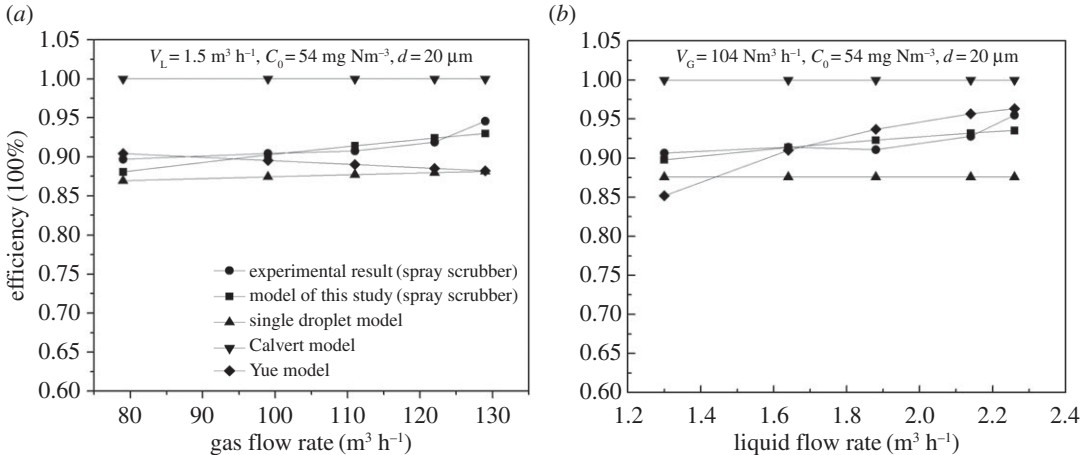

**Figure 3.** Effect of operating parameters on dust removal efficiency in spray scrubber: (a) flow rate of gas and (b) flow rate of slurry.

### 4.1.2. Dust removal model for spray scrubber

Figures 2 and 3 show the predicted collection efficiency by the single droplet equation (2.1), droplets swarm models by Calvert equation (2.12) and Yue equation (2.13) together with the experimental data. Accordingly, the single droplet model can only predict the relationship between the efficiency and the dust particle diameter, and the predicted collection efficiency by the single droplet model is between 88.37% and 88.9%, which are smaller than the actual collection efficiency.

The predicted collection efficiency using the Calvert model is close to 100% except for PM1. Although the predicted tendencies of $V_G$ and $d$ on dedusting efficiency by the Yue model are consistent with the experimental patterns, it is not accurate for $C_0$ and $V_G$. Such deviations for the Calvert and Yue models are mainly due to the integral analysis method, which cannot effectively reflect the dust removal efficiency by multiple droplets.

In a spray scrubber, dust particles come into contact with the sprayed droplets. The parameters related to the properties of dust particles and droplets can influence the dust removal efficiency. The model was established considering the following parameters: (1) operational parameters of the WFGD system such as $V_G$ and $V_L$, (2) dust parameters such as $C_0$ (mg m$^{-3}$) and the density of SiO$_2$ particle $\rho_P$, (3) parameters of scrubbers such as diameter of scrubber $R_t$ and height of scrubber $H$, (4) physical parameters such as viscosity of gas $\mu_g$, viscosity of liquid $\mu_l$, gas density $\rho_G$, density of liquid droplet $\rho_L$ and gravitational acceleration $g$. Their values during the experiment are listed in table 1.

The dust removal efficiency $\eta_{sp}$ based on $\eta_P$ for the single droplet model is given as

$$\eta_{sp} = \eta_P \times f[V_L, V_G, R_t, H, \rho_P, \rho_G, \rho_L, C_0, \mu_g, \mu_L, g]. \tag{4.2}$$

**Table 1.** The values of the parameters.

| parameters | value |
|---|---|
| gas density, $\rho_G$ | 1.15 (kg m$^{-3}$) |
| density of liquid particle, $\rho_L$ | $\rho_L = 1100$ (kg m$^{-3}$) |
| density of SiO$_2$ particle, $\rho_p$ | $\rho_p = 2200$ (kg m$^{-3}$) |
| viscosity of gas, $\mu_g$ | $\mu_g = 1.82 \times 10^{-5}$ (Pa·s) |
| viscosity of liquid, $\mu_l$ | $\mu_l = 8.39 \times 10^{-4}$ (Pa·s) |
| diameter of scrubber, $R_t$ | 0.15 m |
| gravitational acceleration, $g$ | 9.81 (m s$^{-2}$) |

The dimensional analysis was used to reduce the number of independent variables using the Buckingham π-theorem [30] as follows:

$$\eta_{sp} = k_0 \eta_p \left(\frac{R_t V_G \rho_G}{\mu_g \times A}\right)^a \left(\frac{R_t V_L \rho_L}{\mu_l \times A}\right)^b \left(\frac{c_0}{\rho_G}\right)^d \left(\frac{H}{R_t}\right)^e,$$ (4.3)

where $A$ is the cross-sectional area of the scrubber (m$^2$). $A$ is equal to $Rt^2$ and $e$ with the value of 0.485 [30]. The other exponents in equation (4.3) were obtained using the multiple linear regression analysis as follows:

$$a = 0.079; \quad b = 0.075; \quad d = -0.02;$$

Consequently, the equation for the total dust removal efficiency for a spray scrubber was obtained:

$$\eta_{sp} = \frac{0.739 \times \eta_p V_G^{0.0831} V_L^{0.0791} R_t^{0.1659} \rho_G^{0.0868} \rho_l^{0.0791} H^{0.485}}{\mu_g^{0.0831} \mu_l^{0.0791} A^{0.1622} C_0^{0.0037}}.$$ (4.4)

The exponents of parameters in this model are different from those of the Swarm model [30] because the Swarm model was used for the multistage sieve-tray column without considering $\eta_P$. Compared with the models represented by equation (2.12) and equation (2.13), the novel model equation (4.4) is based on the theoretical dust removal efficiency $\eta_P$ and revised by other running parameters. Meanwhile, the new empirical model introduced many new parameters $R_t$, $\rho_G$, $\rho_L$, $C_0$, $\mu_g$ and $\mu_L$, some of which are important for the dust removal, for example, inlet particles concentration [5,6]. A more comprehensive model considering the operating parameters for both dust and WFGD systems was given and will have a practical significance to evaluate the dust removal ability of WFGD systems. When the $n_r$ was less than 2.67, the absolute deviations between the results predicted by equation (4.4) and the experimental data are 0.74%, 0.91%, 3.38% and 0.92%, respectively, for the experimental conditions of gas flow rate, liquid flow rate, dust diameter and inlet concentration. All the deviations are less than 5%, indicating that the established model can better define the influence of operational parameters on dedusting efficiency.

## 4.2. Experimental and modelling results of dust removal efficiency for sieve-tray spray scrubber

### 4.2.1. Experimental result

The application of the WFGD systems with the sieve tray could improve the dedusting efficiency [5,21,23,40]. The study found that a large slurry and flue gas flow rates increased the dust removal efficiency. An increase in the inlet fly-ash concentration and the column height can increase the removal efficiency [30], while the flue gas flow rate did not have an effect [31]. However, the study on the influence of parameters of the sieve tray on the dust removal efficiency is limited.

#### 4.2.1.1. Effect of dust parameters

The influence of the dust particle diameter on the dust removal efficiency is shown in figure 4a. As shown in figure 4, the dust removal efficiency increased from 14.8% to 99% when particle diameter increased from 1 to 20 µm. This is consistent with the results of a multiple sieve-plate column which showed that the dust collection efficiencies were approximately 20%, 30% and 95% for the dust diameters of 0.6 µm, 1.39 µm

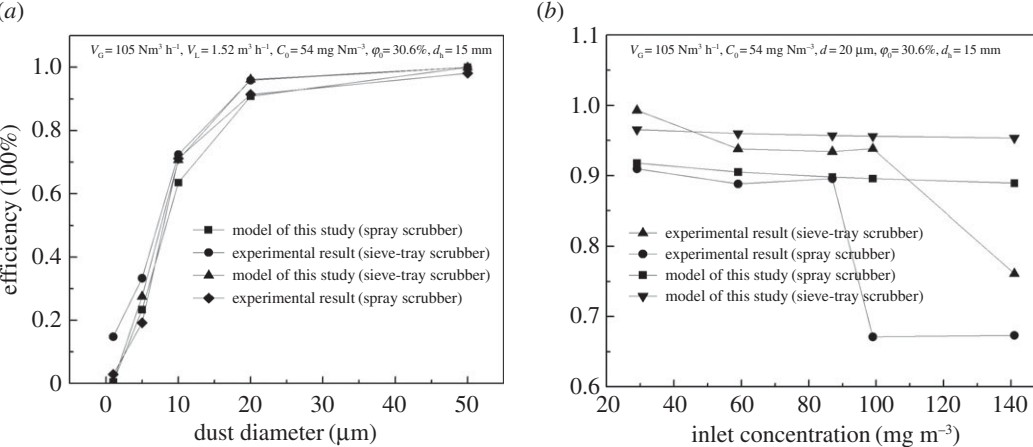

**Figure 4.** Effect of particle parameters on dust removal efficiency in sieve-tray spray scrubber: (*a*) dust diameter and (*b*) inlet dust concentration.

and 10 μm, respectively [26]. The result shows that the WFGD system with a sieve-tray scrubber could easily remove the particles larger than 20 μm, and the efficiency dropped sharply with the decreasing particle diameter. Additionally, the dust removal efficiencies of the sieve-tray spray scrubber were higher than those of the spray scrubber (figure 4*a*), which is consistent with test results that the dedusting efficiency was between 63% and 84% for the sieve-tray spray scrubber, while only between 6% and 69% for the spray scrubber in different commercial power plants [25]. The results illustrate that the synergistic dedusting efficiency under different dust diameter of the WFGD system with a sieve-tray spray scrubber is similar to that of other wet dedusting methods, although the efficiencies were enhanced.

According to figure 4*b*, the dust removal efficiency is negatively correlated with the total dust concentration when the inlet dust concentrations are below 140 mg m$^{-3}$. The enhanced efficiency was about 10.5% compared with that of a spray scrubber. The absorption capacity of the sieve-tray spray scrubber was 140 mg m$^{-3}$, while $n_r$ was 3.81. The dust removal efficiency of the spray scrubber was high compared with that of the sieve-tray spray scrubber because of the high absorption capacity and $n_r$.

### 4.2.1.2. Effect of operational parameters

As shown in figure 5*a,b*, the dust removal efficiency rose from 80.9% to 94.8%, when the gas flow rate increased from 83 to 136 m$^3$ h$^{-1}$. The correlation is positive because the foam height $h_b$ and inertia collision $s_{tb}$ improved with the increased gas flow rate, resulting in an enhanced overall efficiency of inertia collision efficiency. It is worth noting that the dust removal efficiency could be the opposite with increasing gas flow rate [5] because the demister efficiency decreased and carried more particles and droplets out from the scrubber when the gas flow rate continued to increase. When the slurry flow rate increased from 1.52 to 2.52 m$^3$ h$^{-1}$, the dust removal efficiency improved from 86.8% to 98.5%. The results indicate that a large flow rate can strengthen the dust removal efficiency because $\eta_{SI}$ and foam density $F$ in the foam layer increased, resulting in an increased inertia collision efficiency. A similar pattern was observed in a 600-MW power plant with the sieve-tray spray scrubber [5] when the spray layers were adjusted to illustrate that the flow rate can affect the dust removal efficiency.

### 4.2.1.3. Effect of sieve parameters

The effects of porosity ($\Phi_0$) and pore diameter ($d_h$) of the sieve tray on dust removal efficiency are shown in figure 6*a,b*, respectively. For the pore diameter of 15 mm, the dust removal efficiency decreased from 96.2% to 91.2% when the porosity of the sieve tray increased from 23.55% to 40.82%. Because the porosity is negatively related to foam density, synergistic dedusting efficiency decreased as reported by Taheri [26], since the foam density is a function of the geometry and flow parameters of the scrubber.

The dust removal efficiency decreased from 99.3% to 93.3% when the pore diameter increased from 5 to 25 mm. This is because the bubbles became significantly smaller with the decreasing sieve-tray diameter, leading to an increase in the inertial collision ratio $s_{tb}$. Consequently, the inertia collision efficiency increased.

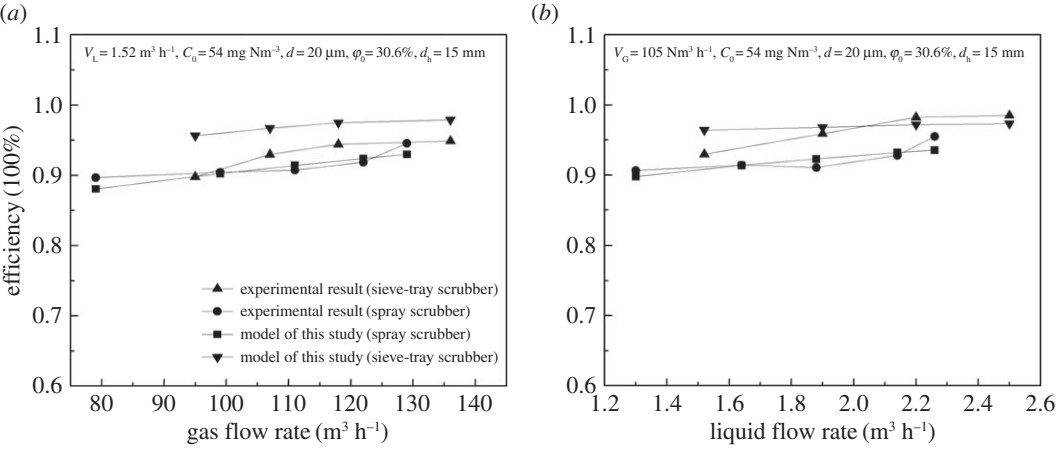

**Figure 5.** Effect of operating parameters on dust removal efficiency in sieve-tray spray scrubber: (a) flow rate of gas and (b) flow rate of slurry.

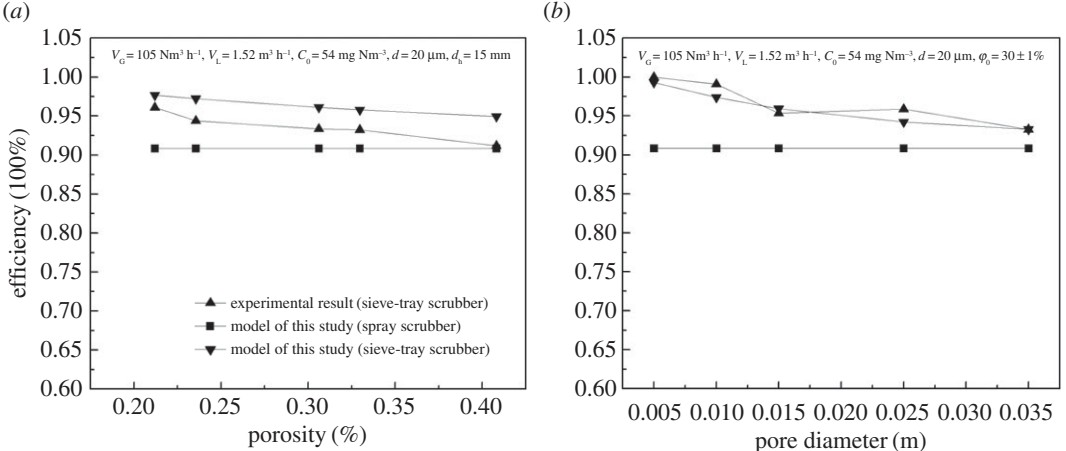

**Figure 6.** Effect of sieve parameters on dust removal efficiency in sieve-tray spray scrubber system: (a) porosity and (b) pore diameter.

### 4.2.2. Dust removal model for sieve-tray spray scrubber

In a sieve-tray spray scrubber, the flue gas and liquid flow spray pass through the holes of the sieve tray to produce the foam layer on the sieve tray, and the foam layer was the main reason for the enhanced dedusting efficiency. The enhanced dedusting efficiency is expressed by equation (4.5), which is the modified Taheri inertia collision model equation (2.14) using a correction factor $K_f$.

$$\eta'_{SI} = 1 - \exp(-40(K_f \cdot F)^2 \cdot S_{tb}). \tag{4.5}$$

$K_f$ is 0.2 obtained through the regression analysis of the experimental data. The revised inertia collision model could be expressed as follows using $K_f = 0.2$,

$$\eta'_{SI} = 1 - \exp(-0.16F^2 \cdot S_{tb}). \tag{4.6}$$

The contribution of the foam layer to the dedusting efficiency, $\eta_s$, can be written as,

$$\eta_s = 1 - (1 - \eta'_{SI})(1 - \eta_{SD}). \tag{4.7}$$

Compared with the spray scrubber, the overall efficiency of the sieve-tray scrubber was higher and the enhancement efficiency of the foam layer $\eta_e$ can be written as

$$\eta_e = \eta_s = 1 - (1 - \eta'_{SI})(1 - \eta_{SD}). \tag{4.8}$$

**Table 2.** The value of $\eta_{SI}'$, $\eta_e$, $\eta_a$ under different operation conditions.

| $V_L$ (m³ h⁻¹) | $V_G$ (m³ h⁻¹) | $d$ (μm) | $C_0$ (mg m⁻³) | $\eta_{SI}'$ (×100%) | $\eta_e$ (×100%) | $\eta_a$ (×100%) |
|---|---|---|---|---|---|---|
| 1.52 | 105 | 1 | 54 | 0.219 | 0.232 | 0.622 |
| 1.52 | 105 | 5 | 54 | 5.338 | 5.343 | 27.506 |
| 1.52 | 105 | 10 | 54 | 19.701 | 19.705 | 70.736 |
| 1.52 | 105 | 20 | 54 | 58.425 | 58.426 | 96.194 |
| 1.52 | 105 | 50 | 54 | 99.585 | 99.585 | 100.00 |
| 1.52 | 79 | 20 | 54 | 43.055 | 43.057 | 93.205 |
| 1.52 | 99 | 20 | 54 | 55.098 | 55.100 | 95.624 |
| 1.52 | 111 | 20 | 54 | 61.602 | 61.603 | 96.703 |
| 1.52 | 122 | 20 | 54 | 67.017 | 67.018 | 97.492 |
| 1.52 | 129 | 20 | 54 | 70.181 | 70.182 | 97.909 |
| 1.3 | 105 | 20 | 54 | 58.425 | 58.426 | 95.754 |
| 1.64 | 105 | 20 | 54 | 58.425 | 58.426 | 96.410 |
| 1.88 | 105 | 20 | 54 | 58.425 | 58.426 | 96.801 |
| 2.14 | 105 | 20 | 54 | 58.425 | 58.427 | 97.176 |
| 2.25 | 105 | 20 | 54 | 58.425 | 58.427 | 97.322 |
| 1.5 | 104 | 20 | 141 | 57.881 | 57.882 | 95.340 |
| 1.5 | 104 | 20 | 99 | 57.881 | 57.882 | 95.606 |
| 1.5 | 104 | 20 | 87 | 57.881 | 57.882 | 95.703 |
| 1.5 | 104 | 20 | 59 | 57.881 | 57.882 | 95.998 |
| 1.5 | 104 | 20 | 29 | 57.881 | 57.882 | 96.544 |

Then, the total dust removal efficiency of sieve-tray spray scrubber is given as [41]

$$\eta_a = 1 - (1 - \eta_e)(1 - \eta_{SP}). \tag{4.9}$$

The value of $\eta_{SI}'$, $\eta_e$ and $\eta_a$ under different operation conditions is shown in table 2. $\eta_{SI}'$ is very important for the total dust removal efficiency of $\eta_a$ in sieve-tray spray scrubbers and is dominant for $\eta_e$; the results illustrated that the inertia collision is the main mechanism for the fourth layer in the sieve-tray scrubber. The dedusting efficiency of sieve-tray spray scrubbers is a combination efficiency contributed by the spray layer and foam layer, and the enhanced dust removal efficiency of sieve trays could be expressed by introducing a strength coefficient to an inertial collision model.

The total dust removal efficiency of $\eta_a$ predicted using equation (4.9) is in good agreement with the experimental data, as shown in figures 4–6. When the $n_r$ is less than 3.81, the absolute deviations between the results predicted by equation (4.9) and the experimental data are 3.91%, 1.65%, 4.39%, 2.41%, 0.92% and 2.69%, respectively, for the experiment conditions of gas flow rate, liquid flow rate, dust diameter, inlet concentration, pore diameter and porosity. The deviation values are less than 5%, suggesting that the results are useful and acceptable for engineering application. The dust removal efficiency in a spray scrubber with a sieve tray was approximately 1.1–10.6% higher than that in the spray scrubber for the same experimental conditions. The results are similar to the enhanced $SO_2$ efficiency of 2–15% in the sieve-tray scrubber experiments [14]. Sieve-tray scrubbers can enhance the $SO_2$ and dust removal at the same level because the enhancement effect is due to the foam layer formed in sieve-tray spray scrubbers.

# 5. Conclusion

This study investigated the synergistic dedusting of the spray scrubber and sieve-tray spray scrubber using the experimental and modelling approaches. The study found that the dust removal efficiency increased

with the increase in dust diameter, gas flow rate and liquid flow rate, and decreased with the increase in the inlet dust concentration, pore diameter and porosity of sieve tray. The dust removal efficiency in a spray scrubber with a sieve tray was approximately 1.1–10.6% higher than that of the spray scrubber for the same experimental conditions. A novel droplets swarm model equation (4.4) was developed in this study based on parameters influencing the dedusting efficiency, including dust diameter, inlet concentration, the flow rate of flue gas and slurry of limestone/gypsum and scrubber dimension. Dimensional analysis was used to simplify the model. A mathematical model equation (4.5)–(4.9) was developed to describe the dust removal effect of sieve-tray spray scrubbers by combining the droplets swarm model for a spray scrubber and the modified foam-based dust removal model equation (4.8). The results simulated using the mathematical models were consistent with the experimental results obtained at various conditions.

# Nomenclature

| | |
|---|---|
| $A$ (m$^2$) | cross-sectional area of the scrubber |
| $C_0$ (mg m$^{-3}$) | inlet dust concentration |
| $C_I$ | correction factor of inertia collision |
| $C_D$ | correction factor of diffusion |
| $C_{out}$ (mg m$^{-3}$) | the outlet dust concentration |
| $C_G$ | correction factor of gravity |
| $C_S$ | correction factor of Stb |
| $C_f$ | foam density in equation (2.15) |
| $d_d$ (m) | droplet diameter |
| $d$ (m) | dust diameter |
| $d_h$ (m) | pore diameter |
| $d_s$ (m) | thickness of tray |
| $d_b$ (m) | bubble diameter |
| $D$ (m$^2$ s$^{-1}$) | diffusion coefficient of particle |
| $F$ (%) | mean foam density |
| $g$ (m s$^{-2}$) | gravitational acceleration |
| $K$ | Boltzmann's constant |
| $k_0, k_f$ | constant |
| $K_P$ | constant of inertial collision frequency |
| $K_f$ | correction factor of inertia collision in the bubble |
| $h_b$ (cm) | height of foam layer |
| $H$ (m) | height of absorber |
| $M_I$ (m) | mass quality of the dust |
| $M_D$ (m) | mass difference before and after collecting dust |
| $n_r$ | the ratio number of particles to droplets |
| $P_e$ | Peclet number |
| $r_b$ (m) | bubble radius in equation (2.19) |
| $R_t$ (m) | scrubber diameter |
| $R$ | intercept coefficient |
| $R_{eD}$ | droplet Reynolds number |
| $R_{eb}$ | sieve-tray Reynolds number |
| Sc | Schmidt number |
| $S_{tb}$ | inertia collision factor |

(Continued.)

| | |
|---|---|
| $T$ (K) | temperature |
| $u$ (m s$^{-1}$) | relative velocity of the gas-liquid |
| $u_g$ (m s$^{-1}$) | gas velocity |
| $u_l$ (m s$^{-1}$) | liquid velocity |
| $u_b$ (m s$^{-1}$) | rising velocity of bubble |
| $u_h$ (m s$^{-1}$) | gas velocity in the hole |
| $V_l$ (m$^3$) | sample volume |
| $V_G$ (m$^3$/h$^{-1}$) | gas flow rate |
| $V_L$ (m$^3$/h$^{-1}$) | slurry flow rate |
| $\rho_p$ (kg/m$^{-3}$) | particle density |
| $\rho_g$ (kg/m$^{-3}$) | gas density |
| $\rho_l$ (kg/m$^{-3}$) | slurry density |
| $\mu_g$ (Pa $\cdot$ s) | gas viscosity coefficient |
| $\mu_l$ (Pa $\cdot$ s) | liquid viscosity coefficient |
| $\eta_a$ (%) | overall efficiency of sieve-tray spray scrubber |
| $\eta_l$ (%) | inertia collision efficiency of single droplet |
| $\eta_R$ (%) | intercept efficiency of single droplet |
| $\eta_D$ (%) | diffusion efficiency of single droplet |
| $\eta_G$ (%) | gravity sedimentation efficiency of single droplet |
| $\eta_P$ (%) | overall efficiency of single droplet |
| $\eta_e$ (%) | enhancement efficiency of foam layer |
| $\eta_S$ (%) | theoretical efficiency of foam layer |
| $\eta_{SD}$ (%) | diffusion efficiency of foam layer |
| $\eta_{Sl}$ (%) | inertia collision efficiency of foam layer |
| $\eta'_{Sl}$ (%) | revised inertia collision efficiency of foam layer |
| $\eta_{SP}$ (%) | overall efficiency of spray scrubber |
| $\varphi_0$ | sieve porosity |

Ethics. There were no humans or animals used in the experiments. The samples were acquired with appropriate ethical approval.

Data accessibility. The data supporting the paper are included in the paper. We have no more data.

Authors' contributions. Q.W. and M.G. designed the study and revised. M.G. and Y.D. supervised and directed the project. Q.W. and H. Z. performed the experiments. Q.W. wrote the manuscript. M.G. and Y.D. revised the manuscript. All authors commented on the manuscript and gave final approval for publication.

Competing interests. We declare we have no competing interests.

Funding. This work is financially supported by China Chongqing Science and Technology Commission Projects (grant no. cstc2017jcyj-yszx0012 and cstc2018jcyj-yszx0016)

Acknowledgements. The authors gratefully acknowledge the State Key Laboratory of Coal Mine Disaster Dynamics and Control for providing experimental conditions. We are also thankful for the senior engineer J. Yu gave important advices for the revised article.

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
