## [Reviewer comments · Royal Society Open Science]

Review History

RSOS-181696.R0 (Original submission)

Review form: Reviewer 1

Is the manuscript scientifically sound in its present form?

Yes

Are the interpretations and conclusions justified by the results?

Yes

Is the language acceptable?

Yes

Is it clear how to access all supporting data?

Yes

Do you have any ethical concerns with this paper?

No

Have you any concerns about statistical analyses in this paper?

No

Recommendation?

Major revision is needed (please make suggestions in comments)

Comments to the Author(s)

This paper reports the synergistic dust removal by the WFGD system, which has a certain practical significance. Besides, the following contents need to be explained further:

1. It is not mentioned in this paper that the effect of temperature as well as pH to removal efficiency. In addition, please explain how to keep pH of the solution in a constant range?
2. It is recommended to add the effect of the liquid gas flow rate ratio (liquid-gas ratio) on the removal efficiency in the sections 4.1.1.2 and 4.2.1.2, which will be more convinced.
3. Please explain effect of gas residence time on the removal effect and the liquid level in the reactor?
4. What does "efficiency" specifically means in all charts? The dust removal rate or the desulfurization rate?

Review form: Reviewer 2

Is the manuscript scientifically sound in its present form?

No

Are the interpretations and conclusions justified by the results?

No

Is the language acceptable?

No

Is it clear how to access all supporting data?

No

Do you have any ethical concerns with this paper?

No

Have you any concerns about statistical analyses in this paper?

No

Recommendation?

Major revision is needed (please make suggestions in comments)

Comments to the Author(s)

The work deals with evaluating the dust removal efficiency in a wet SO₂ removal system as a complementary removal system to the traditional electrostatic precipitator. The work is not easy to follow since the theory is later not connected to the experimental part. Furthermore, the model use must be clearly presented due to the fact that it is based on lab scale experiments, in spite of the use of dimensionless numbers.

- 1.- There are two removal mechanisms, nsp and ns. But later over the development of the empirical model only nsp is considered. Can you elaborate on the reasons? Can you obtain data only for one particular efficiency. How do you decouple the removal efficiency of the foam layer

in the experiments?

2.- The form of eq 27 is not the same as that in the theoretical part, eqs. (12)-(13). Why didn't you use the same form to obtain the fitting?

3.-The models you compare to should be described in the text. A fitting model may be better in a particular case but with less fundamental meaning.

4.- A general comparison on the fitting of the model to experimental data with error bars is needed and a discussion of the usefulness of the correlation.

5.-In the comparison of the results, if the ones provided in the work are use, please cite the equation number in the figures.

Decision letter (RSOS-181696.R0)

11-Mar-2019

Dear Dr Qirong,

The editors assigned to your paper ("Synergistic removal of dust using the wet flue gas desulfurization systems") have now received comments from reviewers. We would like you to revise your paper in accordance with the referee and Associate Editor suggestions which can be found below (not including confidential reports to the Editor). Please note this decision does not guarantee eventual acceptance.

Please submit a copy of your revised paper before 03-Apr-2019. Please note that the revision deadline will expire at 00.00am on this date. If we do not hear from you within this time then it will be assumed that the paper has been withdrawn. In exceptional circumstances, extensions may be possible if agreed with the Editorial Office in advance. We do not allow multiple rounds of revision so we urge you to make every effort to fully address all of the comments at this stage. If deemed necessary by the Editors, your manuscript will be sent back to one or more of the original reviewers for assessment. If the original reviewers are not available, we may invite new reviewers.

If your study uses humans or animals please include details of the ethical approval received, including the name of the committee that granted approval. For human studies please also detail

whether informed consent was obtained. For field studies on animals please include details of all permissions, licences and/or approvals granted to carry out the fieldwork.

- Data accessibility

If you wish to submit your supporting data or code to Dryad (<http://datadryad.org/>), or modify your current submission to dryad, please use the following link:
<http://datadryad.org/submit?journalID=RSOS&manu=RSOS-181696>

- Competing interests

- Authors' contributions

- Acknowledgements

- Funding statement

Kind regards,
Royal Society Open Science Editorial Office
Royal Society Open Science

on behalf of Professor R. Kerry Rowe (Subject Editor)
openscience@royalsociety.org

Associate Editor's comments:

You should fully incorporate and respond to the concerns of the reviewers of your manuscript. As the reviewers comment the manuscript is not always easy to follow, you might benefit from seeking the advice of a language polishing service such as those listed at <https://royalsociety.org/journals/authors/language-polishing/>.

Bear in mind that Royal Society Open Science does not generally allow for multiple rounds of revision. If you are unable to satisfy the reviewers that your manuscript is ready for publication following revision, we may not be able to consider it further.

Comments to Author:

Reviewers' Comments to Author:

Reviewer: 1

Comments to the Author(s)

This paper reports the synergistic dust removal by the WFGD system, which has a certain practical significance. Besides, the following contents need to be explained further:

1. It is not mentioned in this paper that the effect of temperature as well as pH to removal efficiency. In addition, please explain how to keep pH of the solution in a constant range?
2. It is recommended to add the effect of the liquid gas flow rate ratio (liquid-gas ratio) on the removal efficiency in the sections 4.1.1.2 and 4.2.1.2, which will be more convinced.
3. Please explain effect of gas residence time on the removal effect and the liquid level in the reactor?
4. What does "efficiency" specifically means in all charts? The dust removal rate or the desulfurization rate?

Reviewer: 2

Comments to the Author(s)

The work deals with evaluating the dust removal efficiency in a wet SO₂ removal system as a complementary removal system to the traditional electrostatic precipitator. The work is not easy to follow since the theory is later not connected to the experimental part. Furthermore, the model use must be clearly presented due to the fact that it is based on lab scale experiments, in spite of the use of dimensionless numbers.

- 1.- There are two removal mechanisms, nsp and ns. But later over the development of the empirical model only nsp is considered. Can you elaborate on the reasons? Can you obtain data only for one particular efficiency. How do you decouple the removal efficiency of the foam layer in the experiments?
- 2.- The form of eq 27 is not the same as that in the theoretical part, eqs. (12)-(13). Why didn't you use the same form to obtain the fitting?
- 3.-The models you compare to should be described in the text. A fitting model may be better in a particular case but with less fundamental meaning.
- 4.- A general comparison on the fitting of the model to experimental data with error bars is needed and a discussion of the usefulness of the correlation.
- 5.-In the comparison of the results, if the ones provided in the work are use, please cite the equation number in the figures.

Author's Response to Decision Letter for (RSOS-181696.R0)

See Appendix A.

RSOS-181696.R1 (Revision)

Review form: Reviewer 1

Is the manuscript scientifically sound in its present form?

Yes

Are the interpretations and conclusions justified by the results?

Yes

Is the language acceptable?

Yes

Is it clear how to access all supporting data?

Yes

Do you have any ethical concerns with this paper?

No

Have you any concerns about statistical analyses in this paper?

No

Recommendation?

Accept as is

Comments to the Author(s)

this revised manuscript can be accepted by journal.

Review form: Reviewer 2

Is the manuscript scientifically sound in its present form?

Yes

Are the interpretations and conclusions justified by the results?

No

Is the language acceptable?

Yes

Is it clear how to access all supporting data?

No

Do you have any ethical concerns with this paper?

No

Have you any concerns about statistical analyses in this paper?

No

Recommendation?

Accept with minor revision (please list in comments)

Comments to the Author(s)

The work deals with evaluating the dust removal efficiency in a wet SO₂ removal system as a complementary removal system to the traditional electrostatic precipitator. The work is not easy to follow since the theory is not connected to the experimental part. I will summarise the question What is the point in presenting all the model if in the end you develop an empirical correlation and just fit it o the data.

Typically when you compare experimental vs. model , a figure presenting one against the other is reported, instead of the form of Figures 4 and 5

Decision letter (RSOS-181696.R1)

10-May-2019

Dear Dr qirong:

On behalf of the Editors, I am pleased to inform you that your Manuscript RSOS-181696.R1 entitled "Synergistic removal of dust using the wet flue gas desulfurization systems" has been accepted for publication in Royal Society Open Science subject to minor revision in accordance with the referee suggestions. Please find the referees' comments at the end of this email.

The reviewers and Subject Editor have recommended publication, but also suggest some minor revisions to your manuscript. Therefore, I invite you to respond to the comments and revise your manuscript.

- Ethics statement

- Data accessibility

It is a condition of publication that all supporting data are made available either as supplementary information or preferably in a suitable permanent repository. The data accessibility section should state where the article's supporting data can be accessed. This section should also include details, where possible of where to access other relevant research materials such as statistical tools, protocols, software etc can be accessed. If the data has been deposited in an external repository this section should list the database, accession number and link to the DOI for all data from the article that has been made publicly available. Data sets that have been

deposited in an external repository and have a DOI should also be appropriately cited in the manuscript and included in the reference list.

If you wish to submit your supporting data or code to Dryad (<http://datadryad.org/>), or modify your current submission to dryad, please use the following link:
<http://datadryad.org/submit?journalID=RSOS&manu=RSOS-181696.R1>

- **Competing interests**

- **Authors' contributions**

- **Acknowledgements**

- **Funding statement**

Because the schedule for publication is very tight, it is a condition of publication that you submit the revised version of your manuscript before 19-May-2019. Please note that the revision deadline will expire at 00.00am on this date. If you do not think you will be able to meet this date please let me know immediately.

When submitting your revised manuscript, you will be able to respond to the comments made by the referees and upload a file "Response to Referees" in "Section 6 - File Upload". You can use this

to document any changes you make to the original manuscript. In order to expedite the processing of the revised manuscript, please be as specific as possible in your response to the referees.

on behalf of Prof R. Kerry Rowe (Subject Editor)
openscience@royalsociety.org

Associate Editor Comments to Author:

A few remaining modifications are required by the reviewers, please ensure you fully tackle these and provide a suitable response for the reviewers and editors. Good luck!

Reviewer comments to Author:

Reviewer: 2

Comments to the Author(s)

The work deals with evaluating the dust removal efficiency in a wet SO₂ removal system as a complementary removal system to the traditional electrostatic precipitator. The work is not easy to follow since the theory is not connected to the experimental part. I will summarise the question What is the point in presenting all the model if in the end you develop an empirical correlation and just fit it o the data.

Typically when you compare experimental vs. model , a figure presenting one against the other is reported, instead of the form of Figures 4 and 5

Reviewer: 1

Comments to the Author(s)
this revised manuscript can be accepted by journal.

Author's Response to Decision Letter for (RSOS-181696.R1)

See Appendix B.

Decision letter (RSOS-181696.R2)

23-May-2019

Dear Dr qirong,

I am pleased to inform you that your manuscript entitled "Synergistic removal of dust using the wet flue gas desulfurization systems" is now accepted for publication in Royal Society Open Science.

on behalf of Prof R. Kerry Rowe (Subject Editor)
openscience@royalsociety.org

Appendix A

RSOS-181696, titled “Synergistic removal of dust using the wet flue gas desulfurization systems”

Dear Professor R. Kerry Rowe,

Thank you very much for your email on March.12, 2019 about the status of our manuscript. We appreciate your and the two reviewers’ very helpful comments and suggestions, and we have incorporated them in our amendments.

Corresponding answers to the editors comments:

We have made a significant revision to the manuscript according to the reviewers and editors comments. Some points have been highlighted and some have been re-written. All the changes in the revised manuscript are highlighted in yellow, with some of the line numbers listed below in the corresponding answers. Meanwhile, we upload the manuscript and comments file in the system.

Corresponding answers to the reviewers comments:

Reviewer #1

This paper reports the synergistic dust removal by the WFGD system, which has a certain practical significance. Besides, the following contents need to be explained further:

1. It is not mentioned in this paper that the effect of temperature as well as pH to removal efficiency. In addition, please explain how to keep pH of the solution in a constant range?

In commercial power plants, the outlet temperature and PH of WFGD

systems are mostly between 50-60°C and 5.5~6.0 in a stable level , we also controlled them at a stable level in order to study other factors of particles removal, which is usually the changeable running parameters in different power plants.

The PH is controlled by a combined system with a PH meter and peristaltic pump. The PH meter measured the PH value of the slurry in the tank in real time, meanwhile, the peristaltic pump transport the CaO slurry from the MT(Mixing tank) to ST(Slurry tank), if the PH exceed 6.05 in the experiment, the pump was stopped, and was started while the PH dropped to 5.95. In the experience, we controlled the CaO slurry in a stable value, the PH value can also be kept in a stable level, because the consumption of the CaO slurry is stable. It is specially to notice is that the MT and ST should be mixing with a stirrer, otherwise the PH is hardly to be controlled because of the uneven distribution.

2. It is recommended to add the effect of the liquid gas flow rate ratio (liquid-gas ratio) on the removal efficiency in the sections 4.1.1.2 and 4.2.1.2, which will be more convinced.

L/G (liquid-gas ratio) is the ratio of gas flow rate to liquid flow rate. There are usually two ways to change the L/G, the first one is to fix the liquid flow rate and adjust the gas flow rate, another one is to fix the gas flow rate and adjust the liquid flow rate. In this study, we found the change of gas flow rate or slurry flow rate have a different effect in removing particles form WFGD system. In order to avoid confusing **the L/G changed with** the changes of gas flow rate or slurry flow rate, we use them directly to analysis the effect of the particles removal, not the L/G directly.

3. Please explain effect of gas residence time on the removal effect and the liquid level in the reactor?

The gas residence time is equals to the high of the scrubber divided to the gas velocity and the liquid level also can be expressed with the slurry flow rate. We analyzed the gas flow rate and slurry flow rate in the article directly.

4. What does “efficiency” specifically means in all charts? The dust removal rate

or the desulfurization rate?

The dust removal rate, we have pointed out it again in the article and mark out of them in every figures title.

Reviewer #2

The work deals with evaluating the dust removal efficiency in a wet SO₂ removal system as a complementary removal system to the traditional electrostatic precipitator. The work is not easy to follow since the theory is later not connected to the experimental part. Furthermore, the model use must be clearly presented due to the fact that it is based on lab scale experiments, in spite of the use of dimensionless numbers.

1.- There are two removal mechanisms, η_{sp} and η_s . But later over the development of the empirical model only η_{sp} is considered. Can you elaborate on the reasons? Can you obtain data only for one particular efficiency. How do you decouple the removal efficiency of the foam layer in the experiments?

η_{SP} is the overall efficiency of spray scrubber, η_S is theoretical efficiency of foam layer. In the section of 4.2.2, in order to compare the efficiency of the sieve-tray scrubber to spray scrubber, we use the enhanced efficiency η_e to express the enhanced efficiency of the foam in the sieve-tray scrubber. η_e is the revised efficiency of the η_S in the sieve-tray scrubber. We have described the relationship between η_S and η_e in the section 4.2.2 with color.

The overall dust removal efficiency of spray scrubber is only decided by η_{SP} , but the overall dust removal efficiency of sieve-tray scrubber is connected with η_{SP} and η_e (revised of η_S in sieve-tray scrubber).

In the experiments, a sieve-tray is installed in the scrubber (ST shown in Figure 1), which can be installed or take off for the experiments on the sieve-tray scrubber or spray scrubber. The dust removal efficiency of the foam layer is the difference value of them.

2.- The form of eq 27 is not the same as that in the theoretical part, eqs.

(12)-(13). Why didn't you use the same form to obtain the fitting?

Eqs. (12)-(13) is an theoretical model with the method of discrete material balance equation in a cell volume, many parameters that effect the dust removal, such as $R_b, \rho_G, \rho_L, C_0, \mu_g, \mu_L$ are missing, and the results cannot effectively reflect the effect of the parameters in the experiments (we compared the equations results as shown in Fig. 2 and Fig.3). The form of eq 27 is the better way we found to construct the model for simulating the dust removal efficiency for different parameters of the WFGD system.

3.-The models you compare to should be described in the text. A fitting model may be better in a particular case but with less fundamental meaning.

We added the descriptions of the models in the text, and mark out of them with color (lines 221-226).

4.- A general comparison on the fitting of the model to experimental data with error bars is needed and a discussion of the usefulness of the correlation.

We added the comparison of the experimental data to model's results, and marked out of them with color (lines 226-231 for spray scrubber, 304-308 for sieve-tray scrubber).

5.-In the comparison of the results, if the ones provided in the work are use, please cite the equation number in the figures.

We have revised this in the conclusion.

With best regards,

Sincerely yours,

Gu Min, Qirong Wu

Appendix B

RSOS-181696, titled “Synergistic removal of dust using the wet flue gas desulfurization systems”

Dear Professor R. Kerry Rowe,

Thank you very much for your email on May.10, 2019 about acceptance for publication in Royal society Open Science subject to minor revision in accordance with the referee suggestions. We appreciate your and the two reviewers’ very helpful comments and suggestions, and we have incorporated them in our amendments.

We have made a revision to the manuscript according to the reviewer 2. Some points have been highlighted (in green) and some have been re-written. Meanwhile, we upload the manuscript and comments file in the system.

Corresponding answers to the reviewers comments:

Reviewer: 2

Comments to the Author(s)

The work deals with evaluating the dust removal efficiency in a wet SO₂ removal system as a complementary removal system to the traditional electrostatic precipitator. The work is not easy to follow since the theory is not connected to the experimental part. I will summarize the question What is the point in presenting all the model if in the end you develop an empirical correlation and just fit it to the data. Typically when you compare experimental vs. model , a figure presenting one against the other is reported, instead of the form of Figures 4 and 5 .

We add a table in the article to reveal the connection of the theory models to the predicted efficiency, and compares results of the predicted efficiency and experimental data are shown in Figures 4 and 5. In the model analysis, we introduced a corrected parameter into the models to describe the enhanced efficiency (η'_{SI}) of forth layer in the sieve-tray scrubber, which is the dominant for η_e . The dedusting efficiency of sieve-tray spray scrubber is a combination efficiencies contributed by spray layer and enhanced efficiency of foam layer (η_e). Based on of this, the dedusting efficiency of sieve-tray scrubber can be predicted.

We add the descriptions of above in the article with green.

Reviewer: 1

Comments to the Author(s)

this revised manuscript can be accepted by journal.

With best regards,

Sincerely yours,

Gu Min (Corresponding author), Qirong Wu (First author)